# Immediate Effects of the Reverse Plank Exercise on Muscle Thickness and Postural Angle in Individuals with the Forward Shoulder Posture

**DOI:** 10.3390/jfmk7040082

**Published:** 2022-10-07

**Authors:** Dong-Kyun Koo, Seung-Min Nam, Jung-Won Kwon

**Affiliations:** 1Department of Public Health Sciences, Graduate School, Dankook University, Cheonan 31116, Korea; 2Department of Sports Rehabilitation and Exercise Management, Yeungnam University College, Daegu 42415, Korea; 3Department of Physical Therapy, College of Health and Welfare Sciences, Dankook University, Cheonan 31116, Korea

**Keywords:** plank exercise, electromyography, posture correction, ultrasonography

## Abstract

The forward shoulder posture (FSP) results from shoulders being pulled forward by shortened anterior shoulder girdle muscles. The objective of this study was to investigate the short-term effectiveness of the reverse plank exercise on parascapular muscle thickness and forward shoulder angle (FSA) in patients with FSP. Participants were divided into the FSP and non-FSP (NFSP) groups based on the observed angle between the horizontal line of the C7 spinous process and the acromion process. All participants performed a total of five sets of reverse plank exercises at 30 s per set. FSA and muscle thickness of the pectoralis major (PM), serratus anterior (SA), upper trapezius (UT), and lower trapezius (LT) were measured before and after the reverse plank exercise. The muscle thicknesses of the SA and LT, and the FSA, were significantly increased after exercise in the FSP group (*p* < 0.05). Muscle thickness of the PM and UT significantly decreased after the exercise. In the NFSP group, muscle thickness of the LT was significantly increased, and muscle thickness of the PM and UT were significantly reduced after exercise (*p* < 0.05). Upon using between-group analysis, there were significant differences between the FSA, SA, UT, and LT groups (*p* < 0.05). The reverse plank exercise has the short-term benefit of correcting and preventing FSP by increasing SA and LT thickness while decreasing PM and UT thickness. We believe that the reverse plank exercise significantly improved the ability to prevent FSP in FSP-related muscles and was beneficial in achieving optimal postural alignment.

## 1. Introduction

Forward shoulder posture (FSP) is characterized by a protracted, downwardly rotated, and anteriorly tipped scapula with increased cervical lordosis and upper thoracic kyphosis [1]. These alterations consequently increase muscle tension and stress in the neck and shoulder, resulting in pain, numbness, loss of function, and other symptoms of neuromuscular disorders [2,3]. Continuous contractions of the head and neck muscles for long periods gradually contribute to joint misalignment, forward head and shoulder posture, thoracic kyphosis, and loss of the normal cervical lordotic curve [4,5,6]. Risk factors of developing FSP include weakened and lengthened lower trapezius (LT) and serratus anterior (SA) muscles, and shortened, pectoralis minor, pectoralis major (PM), and upper trapezius (UT) muscles [7,8,9,10,11]. FSP can lead to abnormal scapulohumeral rhythm, impingement of rotator cuff tendons, acromioclavicular joint degeneration, bicipital tendonitis, and activation of trigger points [6,12,13].

There are several therapeutic methods used in the management of FSP, such as strengthening the LT and SA muscles [14,15], stretching the pectoralis major and minor [16], wearing a shoulder brace, and making use of kinesiotaping to correct the altered posture [17,18]. These methods can improve the strength of the rhomboid muscles and anterior inclination of the thoracic spine and correct altered scapulohumeral rhythm [19]. To correct malalignment and obtain the ideal posture, it is necessary to use a combination of methods consisting of stretching and strengthening exercises.

Plank exercises are used to strengthen the muscles of the trunk, including the abdominal and back muscles [20]. Additionally, these exercises can be easily modified using various positions to increase muscle strength and level of difficulty. There are many variations of plank exercises, including the front plank, elbow plank, side plank, and reverse plank [21,22]. Among the different plank exercises, the reverse plank focuses on the posterior muscles used in core strength and engages the abdominal muscles when done properly. The reverse plank is performed by extending the shoulder and retracting the scapula while lifting the hips and torso toward the ceiling with straightened joints, forming a straight line with the body and with the feet touching the ground. A previous study reported that the traditional plank engages the anterior stabilizers while the reverse plank engages the posterior stabilizers, including the multifidus, which is easily affected by the direction of gravity due to the supine posture [23]. Additionally, the reverse plank facilitates the strengthening of the muscles of the upper and lower extremities, including the scapular depressors and retractors, elbow extensors, posterior deltoids, gluteus, and hamstrings [24,25,26]. It can also be used as a rehabilitation exercise to improve core and spinal stabilization by engaging both the abdominals and lower back muscles.

Many studies investigated the effectiveness of core exercises, such as the prone and side plank, crunches, bird dogs, and dead bug exercises [27]. Notably, these studies focused on the effect on the core muscles, treatment of patients with low back pain, and improving exercise performance in athletes. Abnormal postures, such as FSP, cause inefficient contraction of the abdominal muscles, resulting in core muscle instability, which in turn leads to altered scapular kinematics and activity of the neck and upper trunk. Although a few studies have reported the effect of the reverse plank on the core muscles [23,28], its effects on the upper trunk muscles associated with FSP have not been established. Thus, the purpose of this study was to investigate the effects of the reverse plank exercise on the postural angle and muscle thickness in individuals with FSP.

## 2. Methods

### 2.1. Participants

In this study, the sample size was set using G-Power software 3.1.3 (Christian-Albrechts-Universität Kiel, Kiel, Germany)(effect size: 0.5; err prob: 0.05; power (1-β err prob): 0.8; Total sample size: 34). Forty subjects were voluntarily recruited from a local university. Participants were divided into the FSP group (when the angle measured between the horizontal line of the C7 spinous process and the acromion process was below 52 degrees) and the non-forward shoulder posture (NFSP) group (when the angle exceeded 52°) [29]. The FSP and NFSP groups consisted of 5 males and 15 females, respectively. Inclusion criteria were as follows: (1) no interventions done for FSP within the last 6 months, (2) no history of balance disorders or musculoskeletal pathologies (such as a history of shoulder surgery or cervical or thoracic fractures), and (3) ability to perform the reverse plank exercise. The purpose and procedures were explained, and the participants provided written informed consent to participate in the study. Ethical approval was obtained from the Institutional Review Board of Dankook University (2019-11-016-002).

### 2.2. Measurements

#### 2.2.1. Forward Shoulder Angle (FSA)

A camera was fixed on a flat table with a tripod using a horizontal level before measurement and was used to take pictures from a distance of 80 cm. The participants were allowed to stand sideways, and the camera was adjusted to be parallel to the C7 vertebrae (Figure 1). The FSA was objectively measured using the GNU Image Manipulation Program 2.10.12 version (Spencer Kimball, Peter Mattis. CA, USA). The FSA was measured three times between the horizontal line of the C7 spinous process and the acromion process [28], and the average of these values was calculated [30].

#### 2.2.2. Ultrasound Imaging

An ultrasound machine with an 8 MHz transducer (SonoAce R7, Samsung Medison Co., Seoul, Korea) was used to measure the muscle thickness of the PM, SA, UT, and LT (Figure 2). For measurement of the PM, the participants were placed in the supine position. The examiner positioned the transducer parallel to the sternum and perpendicular to the clavicle over the third rib. PM thickness was measured by drawing a vertical line from the upper fascia to the lower fascia at a point 1 cm lateral to the origin of the pectoralis minor [31]. SA thickness was measured in the supine position, with the test arm in a neutral horizontal abduction/adduction and at 90° glenohumeral flexion with full elbow extension. The transducer was placed horizontally on the inferior angle of the scapula and was moved laterally to the mid-axillary line. SA thickness was measured by drawing a vertical line from the upper fascia to the upper periosteum over the midpoint of the fifth rib [32]. UT thickness was measured in the prone position with arms at the sides. The ultrasound transducer was placed transverse to the line formed between the 7th cervical vertebra and the acromion at the midpoint. UT thickness was measured by drawing a vertical line from the upper fascia to the lower fascia at the junction of the lower fascia and the supraspinous fossa of the scapula [33]. LT thickness was measured in the prone position, with the head and neck in neutral alignment. Participants placed their arms at the sides, with palms facing upward. The test arm was moved to 120° of abduction and full elbow extension using passive motion. The spinous processes of the 5th thoracic vertebra were palpated, and the transducer was then moved 3 cm laterally to the thickest part of the LT. LT thickness was measured by drawing a vertical line from the upper fascia to the lower fascia at the junction of the lower fascia and the rhomboid major [33].

The transducer was placed vertically to the skin or musculature to minimize the risk of muscle sampling and to ensure replicability. Before obtaining ultrasound imaging, we employed a standardized protocol for measuring muscle thickness to ensure inter-rater reliability [34]. One examiner was trained using the same protocol to perform ultrasound measurements. The examiner was unaware of the group to which the subjects belonged. We used bony landmarks and surface markings to identify a location close to the muscle belly for each muscle. The transducer made the path between the origin and insertion of the muscle. Ultrasound imaging was conducted by placing the transducer over the pre-marked point to measure muscle thicknesses. After the muscles were identified, the examiner slightly retracted the transducer to avoid muscle compression. A sufficient amount of ultrasound gel was used to minimize muscle compression with the transducer head.

#### 2.2.3. Experimental Procedure

The FSA and muscle thickness were measured prior to the reverse plank exercise. Markers were attached to fix the points of measurement of each muscle. To perform the reverse plank exercise, participants were instructed to lift their pelvis vertically off the floor in a supine position, extend their elbows, and align their knees and trunk. Participants were instructed to keep their arms separated, at shoulder-width apart, and maintain a straight line by applying force to the abdomen and hip (Figure 1B). The participants performed a total of five sets, with one set lasting for 30 s and a rest period of 30 s between sets. After the end of the exercise, the same rest period was provided, and then an ultrasound measurement was performed. To improve reproducibility, the ultrasonic measurement position was measured again on the marked area before training. After the entire session was completed, the FSA and muscle thickness were measured to compare changes before and after the reverse plank exercise.

#### 2.2.4. Statistical Analysis

The general characteristics of the participants were analyzed using descriptive statistics. Independent t-tests were performed to analyze the differences between the two groups in terms of age, height, weight, and body mass index (BMI). The paired t-test was used to compare muscle thickness and FSA before and after the reverse plank exercise. The mean difference between the two groups was analyzed using an independent t-test. The null hypothesis was rejected when *p*-values were <0.05. Cohen’s d and 95% confidence intervals (CI) were calculated to determine the within-group and between-group effect sizes. The effect size was categorized as large (≥0.80), moderate (≥0.50), or small (≥0.20) [35].

Thickness data of 40 subjects measured at 2 independent sessions were used for reliability analysis. It was conducted using the SPSS statistical package version 20 (SPSS Inc, Chicago, IL, USA) based on a single-rating, absolute-agreement, two-way random effect model (intraclass correlation coefficient [ICC]_2,1_) [36]. The guidelines given by Currier were used to assess the ICC coefficients: 0.90–0.99: high reliability, 0.80–0.89: good reliability, 0.70–0.79: fair reliability, and ≤0.69: poor reliability [37]. Separate analyses were conducted on each muscle of the images to assess the intraexaminer reliability by comparing before and after the reverse plank exercise. The standard error of the measurements (SEMs) was also calculated to quantify the measurement precision (i.e., SEM = SEM = $s_{x}\sqrt{1-\mathrm{ICC}}$ where $s_{x}$ is the pooled standard deviation). The minimal detectable changes (MDCs) were calculated as SEM × 1.96 × $\sqrt{2}$, which represents the minimal change in muscle thickness that must occur to be 95% confident that a true change occurred [38,39].

## 3. Results

No significant differences in demographic data were observed in terms of age, height, weight, and BMI between the FSP and NFSP groups. Pre-test values of muscle thickness showed no significant differences between the two groups (*p* > 0.05). In addition, there was a significant difference between the FSA of the two groups (*p* < 0.05) (Table 1).

Table 2 shows the comparison of the FSA and muscle thickness of the PM, SA, UT, and LT. Muscle thickness of the SA and LT was significantly increased in the FSP group after the reverse plank exercise (*p* < 0.05). In addition, the muscle thickness of the PM and UT significantly decreased after the reverse plank exercise (*p* < 0.05). In the NFSP group, LT muscle thickness significantly increased after the reverse plank exercise (*p* < 0.05). Furthermore, after the reverse plank exercise, muscle thickness of the PM and UT were significantly reduced (*p* < 0.05). However, the SA muscle thickness was not significantly different after exercise (*p* > 0.05). Regarding the FSA, both the FSP and NFSP groups showed a significant increase after the reverse plank exercise (*p* < 0.05).

Table 3 shows the comparison of changes in the FSA and muscle thickness between the FSP and NFSP groups. In-between group analysis revealed significant differences in FSA, SA, UT, and LT (*p* < 0.05). However, the muscle thickness of the PM was not significantly different between the groups (*p* > 0.05).

Table 4 displays inter-rater reliability analyses for the acquired images of FSA, PM, SA, UT, and LT across all subjects. The ICCs showed values of 0.871 to 0.956 in each muscle and confirmed “good” to “excellent” reliability [40].

## 4. Discussion

To the best of our knowledge, this is the first study to examine changes in the muscle balance of the upper trunk after short-term intervention with reverse plank exercise. The main findings of this study were as follows: (1) the FSA after the reverse plank exercise was significantly increased in the FSP group; (2) the SA and LT thickness was significantly increased in the FSP group, whereas the PM and UT were significantly decreased after the reverse plank exercise; (3) changes in FSA, SA, LT, and UT thickness were significantly greater in the FSP group than in the NFSP group. These results indicate that the reverse plank exercise could help improve the alignment of the FSP and muscle balance of the upper trunk.

Previous researchers have reported that the FSP is caused by a muscle imbalance of the upper trunk, which may be caused by weakness in SA and LT, tightness in the PM and UT, greater thoracic kyphosis, and scapular malalignment [41]. Muscle length changes in the FSP could lead to alterations in the scapular and glenohumeral kinematics [9,42], and for this reason, they are often the focus of therapeutic exercises for scapular rehabilitation and of training programs [42]. In the present study, the degree of FSP was significantly lower after the reverse plank exercise in the FSP group. Our findings are consistent with those of previous studies, although the therapeutic interventions used were different [43,44,45]. Kluemper et al. reported that stretching the anterior shoulder muscles (pectoralis major and minor) and strengthening posterior shoulder muscles (scapular retractors and LT) reduced FSP following a 6-week training program [43]. Lynch et al.’s study showed decreased FSP following an 8-week exercise program, which included stretching of the shortened levator scapula, pectoralis, and UT and strengthening of the lengthened middle trapezius, LT, and SA [44]. Additionally, Roddey et al. reported that a 2-week pectoralis stretching program improved FSP [45]. In our study, successful correction of the FSP may be due to a reverse plank posture, which combined the stretching of the anterior muscle group with the concentric contraction of the posterior muscle group.

The reverse plank selectively activates the back muscles and posterior stabilizers that are easily affected by the direction of gravity due to the supine position and stretches the chest and anterior shoulder muscles. Thus, our results indicate that the reverse plank may be more effective in improving FSP due to the reverse plank posture, which induces posterior tilting in the sagittal plane and external rotation in the transverse plane of the scapula.

In the present study, while muscle thickness for SA and LT significantly increased in the FSP group, the muscle thickness for PM and UT significantly decreased after the reverse plank exercise. Increased muscle thickness indicates shortening of the SA and LT, whereas decreased muscle thickness indicates lengthening of the PM and UT. Changes in muscle thickness that are quantitatively measured using ultrasonography represent relaxed and contracted states of muscles, which are correlated with muscle force [46,47]. Additionally, previous researchers have reported a high correlation between muscle activity and thickness [48,49,50]. Kinematic alterations in scapular motion due to long periods of FSP are linked to a decrease in SA and LT thickness, an increase in PM and UT muscle activity, or muscle imbalances of coupling forces [51,52,53].

There are several possible explanations for the observed changes in muscle thickness. After the reverse plank, muscle activity of the SA and LT may have increased due to decreased UT activity, which compensates for the relatively decreased upward rotators. Scapular external rotation and posterior tilting after the reverse plank may have also increased because of the lengthened PM. This was confirmed by the decreased degree of FSP, and reduced passive tension of the PM may have increased activation of the SA and LT as well. Muscle imbalance between the weakened SA and LT and the tightened PM and UT may also have improved due to the reverse plank. This suggests that alterations in muscle tension contribute to greater force production. Thus, the reverse plank may be sufficient to immediately alter muscle force generation and scapular kinematics in patients with FSP.

However, there was no significant difference in the muscle thickness of the SA in the NFSP group after the reverse plank. The SA has been described as the prime mover of the scapula and is the key contributor to normal and abnormal scapular motion [54]. Additionally, when increased thoracic kyphosis decreases scapular external rotation and posterior tilting, SA activation is consequently changed. Thus, scapular kinematics problems such as FSP are affected by alterations in the SA. The NFSP group has sufficient SA stability to align normal postures, so it was considered that there was no change after the reverse plank.

We also found that changes in SA, LT, and UT thickness were significantly greater in the FSP group than in the NFSP group. These results indicate that the reverse plank contributes to the stretching and strengthening of muscles in patients with FSP compared to patients without NFSP, which is consistent with the findings reported in previous studies [43,55,56]. Consequently, we believe that the reverse plank performed in the present study significantly changed the scapular kinematics due to the stretching and strengthening effect in patients with FSP. However, there was no significant difference in the PM thickness between the two groups. Umehara et al. reported that the PM has variations in lengthening according to the stretching position used, such as shoulder abduction, external rotation, and pelvic rotation [57]. They suggested that the clavicular region of the PM seemed to be effectively stretched below 90° of shoulder abduction with external rotation, whereas the sternal region was effectively stretched at approximately 90° abduction with external rotation. Stegink-Jansen et al. also found that the clavicular and sternal fibers of the PM lengthened differently depending on the stretching position [58]. They showed that the combined motion of a shoulder extension with an external rotation increased the strain of the PM. However, the reverse plank posture used in this study was set below 90° of shoulder abduction with internal rotation to prevent excessive shoulder impingement and hyperextension of the elbow joint. Thus, it seems that the reverse plank posture with lack of shoulder external rotation leads to an insufficient stretch of the PM, which may show no differences between the two groups.

The present study confirms that the reverse plank may result in increased muscle contraction of the SA and LT combined with inhibition of the PM and UT, suggesting that the reverse plank provides immediate correction of the FSP. However, this study has several limitations. First, it is difficult to generalize the results obtained since the population of the participants is composed only of participants between the age of 20 and 30. In addition, it is important to consider the participants because the glenohumeral instability test, which can be affected during the reverse plank, was not performed as part of the participants’ selection criteria. Second, this study focused on the immediate effects of the reverse plank; however, it is also necessary to examine the long-term effects of the exercise. Additional studies are required to determine the persistence of the effects of the reverse plank. In particular, a cohort study is required to examine potential structural changes to the ligament and capsule caused by long-term FSP. Third, future studies need to measure muscle activity since the researchers focused on the effect of treatment only through changes observed in muscle thickness. Furthermore, when maintaining posture during a reverse plank, it is necessary to monitor the change in muscle activity due to the presence or absence of FSP. Finally, the researchers did not measure the muscle thickness of the pectoralis minor in this study because previous reliability and validity studies of muscle morphology measurements of the pectoralis minor using ultrasonography have been insufficient. Additionally, the muscle fascia of the pectoralis minor could not be found because it was too unclear and small to measure. Thus, the researchers chose the PM instead of the pectoralis minor because PM muscle strain occurs in the reverse plank posture with shoulder extension, horizontal abduction, and scapular external rotation. However, the pectoralis minor is an important muscle involved in scapular kinematics, responsible for increases in scapular protraction and anterior tilting, and plays an important role in muscle imbalances associated with FSP [59]. Further studies should consider measuring pectoralis minor thickness for a better understanding of scapular kinematics and muscle imbalances associated with FSP after the reverse plank exercise.

In conclusion, the reverse plank exercise has a short-term effect on the correction of FSP, which results in increased SA and LT thickness and decreased PM and UT thickness. The reverse plank exercise is practical and useful for patients with FSP and allows them to stretch and strengthen muscles concurrently, making use of their weight without requiring tools. We would like to present this exercise as a new method of correcting FSP.

## Figures and Tables

**Figure 1 jfmk-07-00082-f001:**
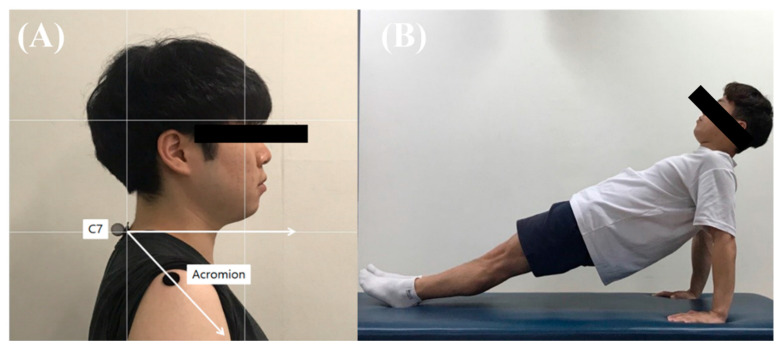
(**A**) Adhesive marker placement and postural angles (**B**) Reverse plank exercise position.

**Figure 2 jfmk-07-00082-f002:**
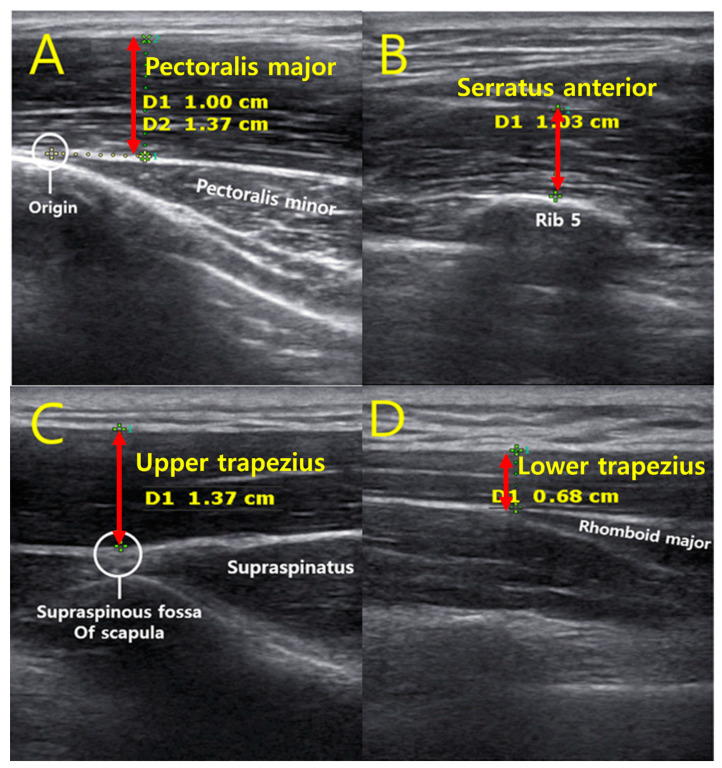
Ultrasound images of the anatomical landmarks and thickness of each muscle; (**A**) Pectoralis major, (**B**) Serratus anterior, (**C**) Upper trapezius, (**D**) Lower trapezius.

**Table 1 jfmk-07-00082-t001:** General characteristics of the subjects.

	FSP Group	NFSP Group	*p*
Age (year)	21.0 ± 1.74	21.7 ± 1.53	0.254
Height (cm)	164.4 ± 7.96	165.7 ± 6.19	0.568
Weight (kg)	59.3 ± 10.10	57.6 ± 8.22	0.551
BMI (kg/m^2^)	21.8 ± 2.42	20.9 ± 1.83	0.169
FSA (°)	46.97 ± 3.61	59.53 ± 5.13	<0.001 *

Values represent mean ± standard deviation; BMI: body mass index; FSA: forward shoulder angle; FSP: forward shoulder posture; NFSP: non-forward shoulder posture. * *p* < 0.05.

**Table 2 jfmk-07-00082-t002:** Comparison of forward shoulder angle and muscle thickness before and after reverse plank exercise.

FSP Group						NFSP Group				
	Pre	Post	t	*p*	ES (95% CI)	Pre	Post	t	*p*	ES (95% CI)
FSA (°)	46.97 ± 3.61	52.31 ± 3.65	−8.165	<0.001 *	−1.47 (−2.170, −0.772)	59.53 ± 5.13	62.11 ± 5.79	−4.802	<0.001 *	−0.47 (−1.100, 0.157)
PM (cm)	0.92 ± 0.29	0.82 ± 0.27	6.506	<0.001 *	0.36 (−0.268, 0.982)	1.01 ± 0.33	0.94 ± 0.31	3.592	0.002 *	0.22 (−0.403, 0.840)
SA (cm)	0.71 ± 0.17	0.79 ± 0.20	−4.765	<0.001 *	−0.43 (−1.058, 0.196)	0.76 ± 0.21	0.78 ± 0.20	−1.469	0.158	−0.10 (−0.718, 0.523)
UT (cm)	0.90 ± 0.28	0.79 ± 0.24	5.561	<0.001 *	0.42 (−0.205, 1.049)	0.87 ± 0.26	0.81 ± 0.25	3.563	0.002 *	0.24 (−0.387, 0.857)
LT (cm)	0.46 ± 0.17	0.57 ± 0.17	−7.769	<0.001 *	−0.65 (−1.283, −0.011)	0.47 ± 0.14	0.53 ± 0.14	−3.449	0.003 *	−0.43 (−1.055, 0.198)

Values represent mean ± standard deviation; Cohen’s d effect sizes (ES) for each group; PM: pectoralis major; SA: serratus anterior; UT: upper trapezius; LT: lower trapezius; FSA: forward shoulder angle; FSP: forward shoulder posture; NFSP: non-forward shoulder posture. * *p* < 0.05.

**Table 3 jfmk-07-00082-t003:** The changes in the forward shoulder angle and muscle thickness between groups.

	FSP Group	NFSP Group	t	*p*	ES (95% CI)
FSA (°)	5.34 ± 2.92	2.57 ± 2.40	3.270	0.002 *	1.04 (0.376, 1.697)
PM (cm)	−0.11 ± 0.07	−0.08 ± 0.10	−0.964	0.341	−0.35 (−0.972, 0.278)
SA (cm)	0.08 ± 0.08	0.03 ± 0.09	2.117	0.041 *	0.59 (−0.046, 1.220)
UT (cm)	−0.11 ± 0.09	−0.05 ± 0.07	−2.220	0.032 *	−0.74 (−1.385, −0.103)
LT (cm)	0.11 ± 0.06	0.06 ± 0.08	2.211	0.033 *	0.71 (0.068, 1.346)

Values represent mean ± standard deviation; Cohen’s d effect sizes (ES) for each group; PM: pectoralis major; SA: serratus anterior; UT: upper trapezius; LT: lower trapezius; FSA: forward shoulder angle; FSP: forward shoulder posture; NFSP: non-forward shoulder posture. * *p* < 0.05.

**Table 4 jfmk-07-00082-t004:** Intra-rater reliability for assessing the muscle thickness composing the forward shoulder posture in all subjects.

	Pre-Test	Post-Test	ICC_2,1_ (95% CI)	SEM	MDC
FSA (°)	53.25 ± 7.72	57.21 ± 6.89	0.890 (0.073–0.968)	2.423	3.426
PM (cm)	0.97 ± 0.31	0.88 ± 0.29	0.956 (0.628–0.987)	0.063	0.090
SA (cm)	0.73 ± 0.19	0.79 ± 0.20	0.928 (0.782–0.969)	0.052	0.073
UT (cm)	0.88 ± 0.27	0.80 ± 0.24	0.950 (0.650–0.984)	0.057	0.081
LT (cm)	0.47 ± 0.15	0.55 ± 0.16	0.871 (0.175–0.959)	0.056	0.078

Values represent mean ± standard deviation; PM: pectoralis major; SA: serratus anterior; UT: upper trapezius; LT: lower trapezius; FSA: forward shoulder angle; ICC: intraclass correlation coefficients; SEM: standard error of measurement; MDC: minimal detectable change (n = 40).

## Data Availability

The data presented in this study are available on request from the corresponding author.

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
