# Peer review of "Immediate Effects of the Reverse Plank Exercise on Muscle Thickness and Postural Angle in Individuals with the Forward Shoulder Posture"

_jfmk, 2022, doi:10.3390/jfmk7040082_

Round 1

Reviewer 1 Report

General comments

The authors aimed to investigate the short-term effectiveness of the reverse plank exercise on parascapular muscle thickness and forward shoulder angle in patients with forward shoulder posture. Findings show this exercise as a new method of correcting forward shoulder posture.

The authors made a very interesting study. The manuscript is well written, and I have no comments to improve the manuscript.

Specific comments

Keywords

Please use keywords other than the title. In order to optimize the search for your article through search engines, I suggest you replace the keywords “forward shoulder posture; reverse plank exercise ”with other relevant ones.

Introduction

The introduction correctly summarizes the background, what is known, what is unknown, the gap in the literature to be filled, and the research objectives.

Methods

The Methods are written in a detailed and reproducible manner. The statistics used are appropriate.

Results

The results are clear and detailed.

Discussion

The discussions correctly interpret the results that were compared in detail with previous studies. The limitations are described. The take-home message is clear.

Reviewer 2 Report

Thank you for the opportunity to review this manuscript, below are my considerations on the article.

Abstract

- What muscle is this “parascapular”?

- Why was the pectoralis major muscle selected and not the pectoralis minor?

- The authors mention that the reverse plank exercise is able to prevent the anterior posture of the shoulders and they did not measure this outcome. I suggest removing this topic.

Introduction

 In the introduction, the authors talk about the pectoralis minor muscle, however, it seems that they evaluated the pectoralis major.

Methods

- What is type of study?

- How was the sample estimated? Was there a calculation to estimate the sample? Is this a convenience sample?

- How was the sample recruited?

- How was the sample divided between the study groups?

- There is a confusion in which muscles were recruited, the text says pectoralis major, at another time pectoralis minor, then goes back to talking about pectoralis major and in figure 2A they mention pectoralis minor. Authors need to standardize throughout the text the muscle that was part of the study.

- The authors mention how the ultrasound identified the muscles and the path of the transducer, I suggest that the authors modify it, citing that the transducer made the path between the origin and insertion of the muscle, mentioning the point of origin and insertion of each muscle.

- Have the evaluators received training for the evaluation? Was it the same rater or more than one rater?

- Did the evaluators know which group the participants they were evaluating were from?

- Was the study evaluated and approved by any Research Ethics Committee? Where are the study ethics statements?

- Has the protocol of the intervention been published in any registration body? What is the protocol registration identification number?

Discussion

- The authors mention that the exercise used is effective to improve FSP alignment, I suggest that the authors be more cautious, considering that it is only a single session.

- The authors mention that they did not evaluate the pectoralis minor, however, in figure 2 A, the photo shows the pectoralis minor, which is a muscle involved in the clinical pathophysiology of the anterior shoulder. The text is very confusing.

Reviewer 3 Report

It is an interesting article in which the authors use ultrasound to assess muscle thickness and relate it to muscle activity in the reverse plank exercise. By focusing exclusively on the image, the global vision of what is happening in the joint is lost. The muscles are not alone, there are more structures. The thickness is not the only thing that changes in the muscle.

In the population, the authors do not indicate the gender distribution in the groups. An imbalance between men and women could change values. Joint instability is associated with hyperlaxity and this situation is more frequent in women. Likewise, the article does not mention this possibility either and speaks exclusively of muscle, without taking into account the other joint structures (capsule, ligaments). The authors should refer to this situation and justify why it is not taken into account.

The authors must also justify why a clinical examination is not performed to determine the presence of glenohumeral instability. In addition, they also do not assess muscle strength or balance. They only do a static study of the shoulder position and assume that the differences are exclusively muscular in origin.

In the method, the authors should indicate the number of explorers who performed the ultrasound studies.

The authors must also explain whether it was the same or a different explorer who performed the second measurement and state that the person who performed the second examination did not have the data from the first, he was "blinded".

In addition, they should indicate the order in which the measurements were taken and the time it took to obtain them, especially the last post-test measurement. Does the time elapsed since the end of the exercise and the measurement of muscle thickness influence? Have the authors done anything to check that this measurement is correct and not influenced by rest?

In the second paragraph of the statistical analysis section, information provided in the first paragraph (SPSSv20) is repeated.

The significant difference between the FSA of the two groups (p < 0.05) (Table 1) is due to the creation of the groups with this criterion. It is not a finding of the study, it is something predictable.

The paragraph of results that begins “Table 4 displays inter-rater reliability…” provides numerical values that are in the table and are redundant.

At the end of the second paragraph of the discussion the authors write: “In our study, successful correction of the FSP may be due to a reverse plank posture, which combined the stretching of the anterior muscle group with the strengthening of the posterior muscle group”. I think that with a single session it is not possible to speak of "strengthening", it would be more correct to speak of concentric contraction. In a single session the muscles do not get stronger, they contract.

In the discussion, the authors opine that "Thus, it seems that the reverse plank posture with lack of shoulder external rotation leads to an insufficient stretch of the PM, which may show no differences between the two groups." Could this be associated with an eccentric contraction of the muscle to keep the posture balanced and not just a stretch? Authors should include their reasoning or justification in the discussion.

In the limitations of their work or in suggestions for future studies, the authors should include the use of an electromyographic study while maintaining the posture to determine the activity of each muscle, in addition to its morphology, comparing people with FSP and without FSP.

Another limitation of the work is that it does not answer the question: Do men and women respond the same way?

In the references, the authors must homogenize the use of uppercase and lowercase letters.

Reviewer 4 Report

Minor revision

The article is methodologically well-conducted and both the introduction and discussion are scientifically supported.

1- Please cover the participant's face in figure 1B.

2- Participant section. Please put "[27]." instead of ".[27]".

3- Same for the end of section 2.2.4. and throughout the manuscript.

4-  Please put the reference section according to the journal regulations.

Round 2

Reviewer 2 Report

I thank the authors for the changes in the manuscript, the changes made the text more fluid and more meaningful for readers.

Reviewer 3 Report

The authors have responded appropriately to the comments.